# Divergent wiring of repressive and active chromatin interactions between mouse embryonic and trophoblast lineages

Stefan Schoenfelder [1,2], Borbala Mifsud [3,8], Claire E. Senner[2,4], Christopher D. Todd[5], Stephanie Chrysanthou[2], Elodie Darbo[6,7], Myriam Hemberger [2,4] & Miguel R. Branco [5]

The establishment of the embryonic and trophoblast lineages is a developmental decision underpinned by dramatic differences in the epigenetic landscape of the two compartments. However, it remains unknown how epigenetic information and transcription factor networks map to the 3D arrangement of the genome, which in turn may mediate transcriptional divergence between the two cell lineages. Here, we perform promoter capture Hi-C experiments in mouse trophoblast (TSC) and embryonic (ESC) stem cells to understand how chromatin conformation relates to cell-specific transcriptional programmes. We find that key TSC genes that are kept repressed in ESCs exhibit interactions between H3K27me3-marked regions in ESCs that depend on Polycomb repressive complex 1. Interactions that are prominent in TSCs are enriched for enhancer–gene contacts involving key TSC transcription factors, as well as TET1, which helps to maintain the expression of TSC-relevant genes. Our work shows that the first developmental cell fate decision results in distinct chromatin conformation patterns establishing lineage-specific contexts involving both repressive and active interactions.

[1] Nuclear Dynamics Programme, Babraham Institute, Cambridge CB22 3AT, UK. [2] Epigenetics Programme, Babraham Institute, Cambridge CB22 3AT, UK. [3] William Harvey Research Institute, Barts and the London School of Medicine and Dentistry, QMUL, London EC1M 6BQ, UK. [4] Centre for Trophoblast Research, University of Cambridge, Cambridge CB2 3EG, UK. [5] Blizard Institute, Barts and The London School of Medicine and Dentistry, QMUL, London E1 2AT, UK. [6] Bordeaux Bioinformatics Center, 33076 Bordeaux, France. [7] INSERM ACTION U1218, Institut Bergonié, 33076 Bordeaux, France. [8] Present address: College of Health and Life Sciences, Hamad Bin Khalifa University, Education City, Doha, Qatar. These authors contributed equally: Stefan Schoenfelder, Borbala Mifsud. Correspondence and requests for materials should be addressed to M.H. (email: myriam.hemberger@babraham.ac.uk) or to M.R.B. (email: m.branco@qmul.ac.uk)

Mammalian preimplantation development involves dramatic molecular changes that enable the consolidation of the information carried within the oocyte and sperm into a novel, and fast-evolving, gene expression programme that gives rise to the embryo.

Importantly, during preimplantation development the embryo undergoes the first cell lineage differentiation process, with the inner cell mass (from which all embryonic tissues are derived) and trophectoderm (giving rise solely to extraembryonic tissues) being clearly delineated by the blastocyst stage. This morphological event constitutes a strict separation of cell fate, which is underpinned by the establishment of a strong epigenetic barrier set up by DNA methylation and histone modifications, such as H3K9me3, that prevent trans-differentiation from one cell type to the other[1–5].

Our knowledge of the factors that drive the establishment and maintenance of the trophoblast lineage has greatly benefited from the ability to culture mouse trophoblast stem cells (TSCs) in a stable manner, which retain the full differentiative capacity of the early trophoblast[6]. Together with in vivo studies using mouse mutants, this has allowed for a dissection of the signalling events, transcription factors and epigenetic mechanisms that regulate the trophoblast lineage, maintaining its cellular identity and preventing trans-differentiation into embryonic stem cells (ESCs)[7]. Key transcription factors such as CDX2, EOMES, ELF5, TEAD4 and GATA3, among others, are essential for TSC self-renewal[1,8–11]. These networks are distinctly different from those operating in ESCs, which are prominently made up of OCT4, NANOG, KLF4, ZFP42 and others[12]. Intriguingly, some transcription factors, such as SOX2 and ESRRB, are seemingly pivotal in both cell lineages and stem cell types. However, despite this shared necessity, they operate in diverging protein complexes and bind to largely non-overlapping genomic loci in both stem cell types, which endows them with a cell type-specific function[13,14].

These stem cell type-specific transcriptional networks are associated with dramatic differences in the epigenomic landscape between the trophoblast and the embryonic lineages[15]. Most prominently, the DNA in trophoblast cells is found in a globally hypomethylated state[16–19]. Interestingly, loss of DNA methylation in ESCs enhances trans-differentiation into TSCs, which is due in part to the activation of *Elf5* expression[1]. TSCs are also characterised by overall low levels of the repressive H3K27me3 mark[20], which in ESCs is often found co-localised with the active H3K4me3 mark at developmental genes, creating an epigenetic bivalency[21]. In TSCs, loss of H3K27me3 either enables the expression of lineage-determining genes such as *Cdx2*, or triggers the deposition of alternative repressive marks, namely H3K9me3 and DNA methylation[22,23].

Despite this large body of epigenomic data, it remains unknown how this information is integrated by the three-dimensional (3D) arrangement of the genome to establish a TSC-specific gene expression programme. Detailed studies in multiple cell types, including mouse and human ESCs[24,25], have abundantly demonstrated that linear proximity in the genome cannot predict how localised epigenomic changes will affect gene expression. Namely, *cis*-acting enhancer elements can act on genes lying several hundreds of kilobases (or even megabases) away[26]. Interactions associated with repressive marks can also be detected at long distances within a chromosome or even across chromosomes, as was recently demonstrated for a large Polycomb-driven interaction network between *Hox* gene clusters[27].

Here we investigate the differences in chromosome conformation between the embryonic and extraembryonic lineages, using mouse ESCs and TSCs as models. We integrate these data with epigenomic information and transcription factor binding profiles to uncover key regulatory principles governing the regulation of the TSC-specific transcriptional programme. We find that TSC-specific genes are associated with: (1) repression in ESCs involving Polycomb-dependent interactions and (2) the action of key transcription factors and TET1 at cell-specific gene–enhancer contacts in TSCs.

## Results

**Distinct spatial organisation of TSC and ESC genomes**. We used promoter capture Hi-C (PCHi-C)[24,28] to gain an in-depth view of 3D contacts with gene promoters in mouse TSCs and ESCs. Two replicates of each cell type were performed, yielding 51–75 M mapped valid read pairs per replicate and an average of 1730 chimeric reads per captured promoter fragment (Supplementary Fig. 1a). All analyses were performed at the resolution of single HindIII fragments, with good overall concordance between replicates up to distances of ~300 kb (Supplementary Fig. 1b), although strong longer range interactions were also robustly detected (see examples in Supplementary Fig. 4a). Statistically significant interactions were identified using GOTHiC[29], and filtered to keep only interactions that were above the background significance level for each promoter (see Methods). Sex chromosomes were excluded due to sex differences between the cell lines. This yielded a total of 690,074 and 672,062 interactions (in TSCs and ESCs, respectively) between promoters and non-promoter regions (Fig. 1a). We also detected 152,992 and 201,793 (in TSCs and ESCs, respectively) promoter–promoter interactions using a distinct background model for this purpose (Fig. 1a). To gauge the degree of similarity between the promoter interactomes of TSCs and ESCs, we compared the overlap between these cell lines with that seen between different hematopoietic cell types[30]. The overlap between both stem cell types was smaller than that between any of the hematopoietic lineages, including comparisons between vastly different lymphoid and myeloid cell types (Supplementary Fig. 2a), suggesting that TSCs and ESCs differ substantially in their 3D arrangement of chromatin.

To identify interaction differences between TSCs and ESCs in a stringent manner, we developed a statistical model (see Methods) that performs a direct comparison of the mapped data from both cell types, yielding a list of statistically significant differential interactions. These differential interactions also displayed strong differences in contact frequency in a genome-wide Hi-C dataset from an independently derived ESC line (Supplementary Fig. 2b)[31]. Notably, genes involved in TSC-specific interactions included a number of essential trophoblast-associated genes such as *Tfap2c* (Fig. 1b), *Eomes*, *Tead4*, *Id2* and *Fgfr2*. ESC-specific interactions included various developmental genes such as *Six3*, *Pax1* (Fig. 1b), *Evx1* and *Myocd*, as well as the *Hox* gene clusters (Fig. 1c, Supplementary Fig. 2c). A *Hox* gene interaction network has previously been shown to be driven by the action of Polycomb repressive complex (PRC) 1 in ESCs[27]. In line with reduced PRC activity in TSCs[20], H3K27me3 levels at *Hox* gene clusters were low in TSCs and interactions within each cluster largely reduced in strength or even absent (Fig. 1c, Supplementary Fig. 2c).

These results show that ESCs and TSCs differ dramatically in their spatial organisation of the genome, which may contribute to the distinct cell lineage identity of both stem cell types.

**H3K27me3 interactions in ESCs involve TSC-specific genes**. Given the striking difference in interactions at *Hox* gene clusters, we asked whether interactions between H3K27me3-marked regions in ESCs were generally absent in TSCs. Using publicly available ESC ChIP-seq data, we identified interactions where both ends are marked by the same histone mark (or by CTCF), hereafter called homotypic interactions. We found that a

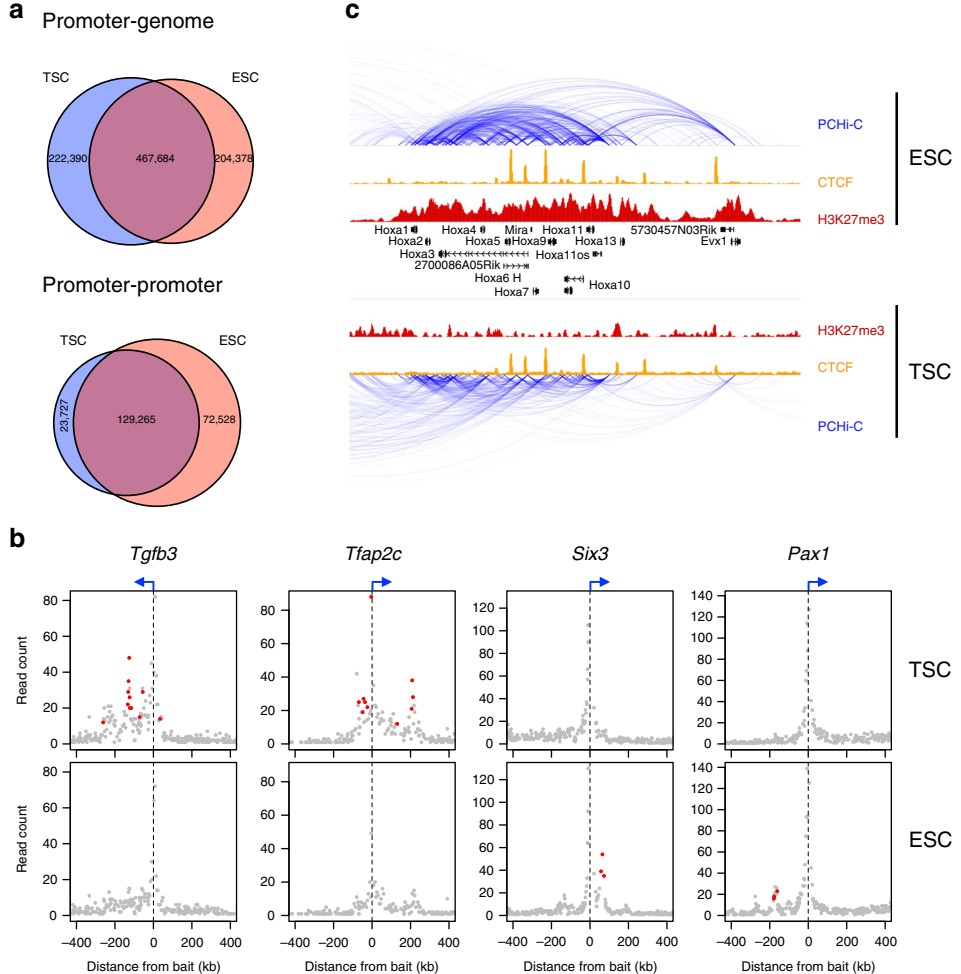

**Fig. 1** Differential interactions between TSCs and ESCs. **a** Venn diagrams comparing the significant interactions from TSC and ESC PCHi-C data, separated into interactions with non-promoter regions (promoter–genome) or with other promoters (promoter–promoter). **b** Examples of differential interactions between TSCs and ESCs. Read counts of fragments interacting with selected promoters are displayed, with differential interactions highlighted in red. **c** Genome browser view of the *Hoxa* gene cluster highlighting a reduction in intra-cluster interactions in TSC cells

disproportionally large amount of H3K27me3 homotypic interactions were ESC-specific ($p < 2.2E-16$, Chi-squared test; Fig. 2a). With the exception of H3K9me3, all other homotypic interactions involved a much smaller fraction of ESC-specific contacts (Fig. 2a). Notably, 29% of the ESC-specific H3K27me3 homotypic interactions were between gene promoters, whereas promoter–promoter interactions accounted for only 3% of H3K9me3 interactions. Thus, ESCs are characterised by a particular enrichment for H3K27me3-associated promoter–promoter interactions that are absent from TSCs. The reverse was not true, as TSC-specific interactions were rarely associated with H3K27me3, involving only five genes.

We then asked whether ESC-specific homotypic interactions were epigenetically distinct from interactions that are shared between the two cell types. Promoters involved in H3K27me3 homotypic interactions in ESCs lacked this mark in TSCs, irrespective of whether these interactions were ESC-specific or not (Fig. 2b). However, we noticed that H3K27me3 peaks associated with ESC-specific interactions were larger (median 901 bp vs. 471 bp for all interactions; Supplementary Fig. 3a) and displayed lower levels of H3K4me3 when compared against all homotypic interactions (Fig. 2b, Supplementary Fig. 3a). Among the shared interactions, even the broader H3K27me3 domains were characterised by lower H3K4me3 (Supplementary Fig. 3a). ESC-specific interactions are therefore characterised by a broad

deposition of H3K27me3 and depletion of H3K4me3. At H3K9me3 homotypic interactions, peaks associated with ESC-specific interactions were indistinguishable from the total pool of H3K9me3 interactions (Fig. 2c, Supplementary Fig. 3b).

We then hypothesised that ESC-specific H3K27me3 interactions maintained repression of TSC genes. Indeed, the cohort of genes associated with ESC-specific H3K27me3 interactions exhibited lower expression levels in ESCs compared to TSCs (Fig. 2d), and this was not due to the broad deposition of H3K27me3 at these genes (Supplementary Fig. 3c). By comparison, no such difference was observed when including all genes involved in H3K27me3 homotypic interactions or genes that are not involved in homotypic interactions (Fig. 2d). This effect was exclusive to H3K27me3-associated interactions, as no significant expression differences were seen for H3K9me3- or CTCF-associated genes (Fig. 2d, Supplementary Fig. 3d). Moreover, the skewed TSC/ESC expression ratio at ESC-specific interactions cannot be explained by promoter H3K27me3 occupancy alone, but strictly required both interacting ends to be marked by H3K27me3 (Supplementary Fig. 3e).

We then focused specifically on TSC-expressed genes that were involved in H3K27me3 homotypic interactions in ESCs. Strikingly, among these we uncovered an ESC-specific long-range interaction between *Gata3* and *Sfmbt2* (Fig. 2e, Supplementary Fig. 4a). *Gata3* encodes a key transcription factor involved in the

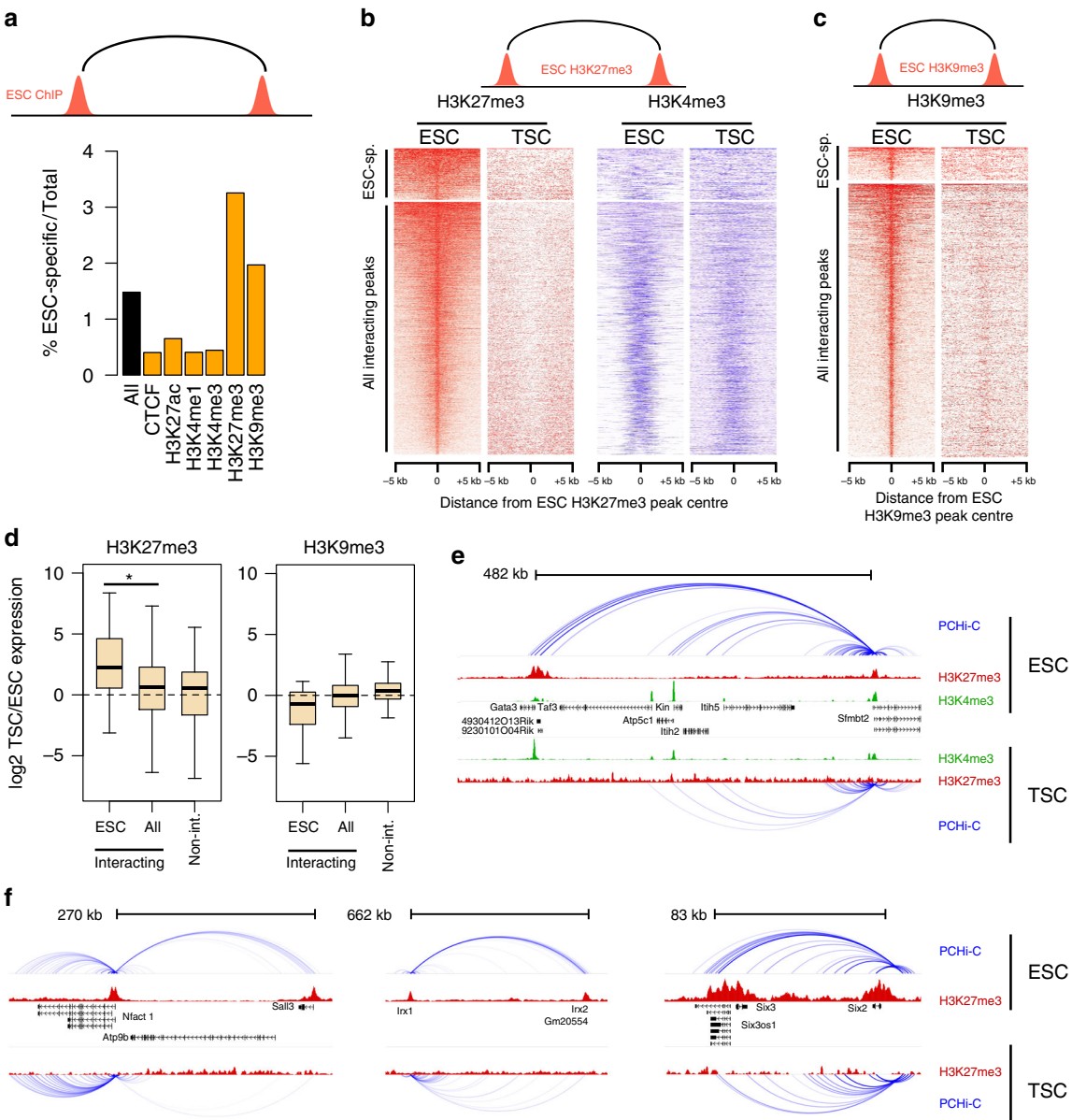

**Fig. 2** H3K27me3 interactions in ESCs involve TSC-specific genes. **a** Percentage of ESC-specific interactions among all interactions (black bar) or homotypic interactions involving the proteins/modifications indicated (orange bars). **b** H3K27me3 and H3K4me3 ChIP-seq signals centred on the peaks involved in ESC H3K27me3 homotypic interactions. Top heatmaps are for ESC-specific interactions only. **c** H3K9me3 ChIP-seq signals centred on the peaks involved in ESC H3K9me3 homotypic interactions. **d** Expression ratio (log2) between TSC and ESCs for genes involved in either H3K27me3 or H3K9me3 homotypic interactions (all or ESC-specific), and compared with genes not involved in homotypic interactions. Boxplot midline represents median, box edges the first and third quartiles, and whisker edges are the last data points within 1.5× the interquartile range. *$p < 0.05$ (t-test). **e**, **f** Examples of H3K27me3 homotypic interactions in ESCs involving TSC-expressed genes

establishment of the trophoblast lineage[11], and *Sfmbt2* is an imprinted gene essential for maintenance of the trophoblast lineage[32]. Other examples of TSC-expressed genes that are involved in such long-range repressive interactions in ESCs included *Nfatc1* and the homeobox genes *Irx1*, *Irx2* and *Six2* (Fig. 2f, Supplementary Fig. 4a). These interactions could also be robustly detected in a genome-wide Hi-C dataset (Supplementary Fig. 4b)[31].

These results suggest that H3K27me3 homotypic interactions, coupled with low H3K4me3 levels, may help maintain a repressed state of trophoblast genes in ESCs.

**ESC-specific H3K27me3 homotypic interactions require PRC1.** Given that interactions within and across *Hox* gene clusters in

ESCs depend on PRC1 (ref. [27]), we asked whether PRC1 is indeed necessary for all the ESC-specific homotypic interactions identified above. For this purpose, we analysed published PCHi-C data from a *Ring1A* knockout (KO) ESC line that also allows for tamoxifen-inducible deletion of *Ring1B*, thus generating *Ring1A/B* double knockout (DKO) cells[27]. We found that, compared to KO cells, DKO cells had substantially fewer of the H3K27me3 ESC-specific interactions that we identified ($p = 2E-10$, proportions test; Fig. 3a), and those that were retained were generally weaker (Fig. 3b). In contrast, there was a similar number of CTCF- and H3K9me3-associated interactions of comparable strength (Fig. 3a, b). ChIP-seq data from the same cell lines[33] showed that all H3K27me3 homotypic interaction sites displayed a loss of H3K27me3 in DKO cells, as well as reduced binding of

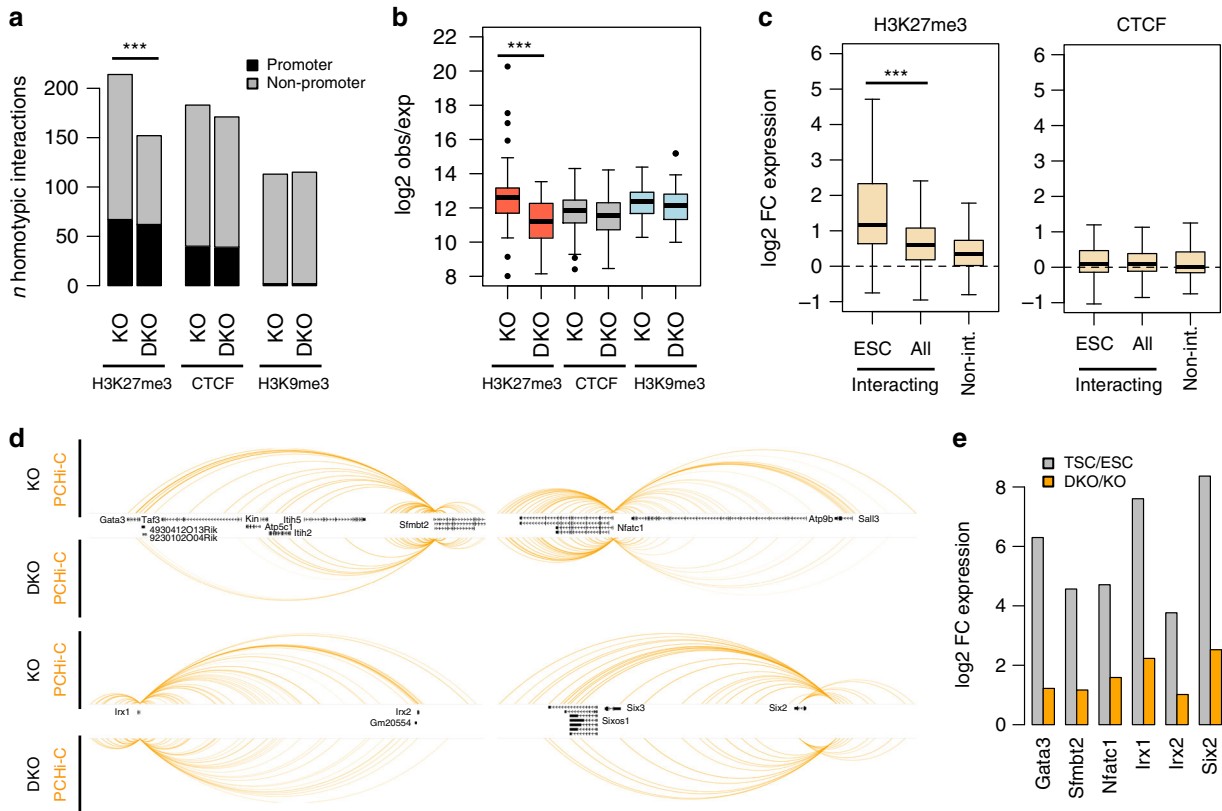

**Fig. 3** ESC-specific H3K27me3 homotypic interactions require PRC1. **a** Number of wild-type homotypic interactions (involving the indicated proteins/modifications) found in *Ring1A* KO or *Ring1A/B* DKO ESCs. **b** Strength of the preserved homotypic interactions (measured as log2 observed/expected PCHi-C signal) in *Ring1A* KO or *Ring1A/B* DKO ESCs. ***$p < 0.0005$ (t-test, corrected for multiple KO-DKO comparisons). **c** Expression ratio (log2) between *Ring1A/B* DKO and *Ring1A* KO ESCs for genes involved in either H3K27me3 or CTCF homotypic interactions (all or ESC-specific), and compared with genes not involved in homotypic interactions. ***$p < 0.0005$ (t-test). **d** Interaction profiles in *Ring1A* KO or *Ring1A/B* DKO ESCs for the regions highlighted in Fig. 2e, f. **e** Expression ratio between *Ring1A/B* DKO and *Ring1A* KO ESCs for TSC-expressed genes involved in the interactions highlighted in **d**. Boxplot midline represents median, box edges the first and third quartiles, and whisker edges are the last data points within 1.5× the interquartile range

PRC1 (RING1B) and PCR2 (EZH2 and SUZ12) proteins (Supplementary Fig. 5a). This reduced PRC binding was even more pronounced when focussing on those loci involved in ESC-specific interactions (Supplementary Fig. 5b).

To test whether loss of interactions in *Ring1A/B* DKO cells was coupled to expression changes, we analysed RNA-seq data from the same cell lines. The expression of genes involved in H3K27me3 homotypic interactions was elevated in DKO cells when compared to KO cells, and this was particularly pronounced for ESC-specific interactions (Fig. 3c). In contrast, expression of genes involved in CTCF-associated interactions remained unchanged (Fig. 3c). Importantly, the upregulation of H3K27me3-associated genes was not restricted to *Hox* genes, but also involved multiple other genes (Supplementary Fig. 5c).

Focusing on the same examples as before (Fig. 2e, f), we found loss or clear weakening of interactions at TSC-expressed genes that were marked by H3K27me3, including *Sfmbt2* and *Gata3* (Fig. 3d, Supplementary Fig. 6). Moreover, all TSC-expressed genes involved in these interactions were upregulated in DKO cells, although they did not reach the same expression levels seen in TSCs (Fig. 3e), presumably due to differences in transcription factor availability. These results indicate that PRC1 is required to establish repressive interactions that are associated with silencing of trophoblast lineage genes in ESCs.

**TSC-specific interactions involve TET1-regulated enhancers.** We then focused on interactions between gene promoters and non-promoter regions. To find out which epigenetic signatures predominate at non-promoter ends of cell-specific interactions, we used the above ChIP-seq data and further defined three groups of enhancer elements: active (H3K4me1 + H3K27ac), poised (H3K4me1 + H3K27me3) and intermediate (H3K4me1 alone). Interactions with CTCF sites, as well as with intermediate and active enhancers, were enriched in TSC-specific interactions (Fig. 4a). This is strikingly distinct from the pattern seen for ESC-specific interactions, which predominantly involve H3K27me3 and poised enhancers (Fig. 4a), similar to what is seen for homotypic interactions. Indeed, 82% of interacting poised enhancers contact at least one promoter marked by H3K27me3 in ESCs. Loss of PRC1 in ESCs is associated with an enhancer switch from poised to active and upregulation of interacting genes[27]. Notably, we found that a subset of ESC poised enhancers (82 out of 490; 17%) are found in the active state in TSCs (Supplementary Fig. 7a, b). These enhancers are associated with higher expression of interacting genes in TSCs (Supplementary Fig. 7c), which include *Cdx2*, *Eomes* and *Dlx3*, suggesting that a number of key TSC enhancers may be kept repressed by Polycomb-driven interactions in ESCs.

To investigate in more detail interactions with active TSC enhancers, we first defined a more stringent set of active enhancers by incorporating chromatin accessibility data (see Methods). As expected, genes interacting with at least one active TSC enhancer displayed higher expression than those with no active enhancer interactions (Supplementary Fig. 8a). These genes also had higher relative expression in TSCs when compared to ESCs, in particular if at least one of those gene–enhancer

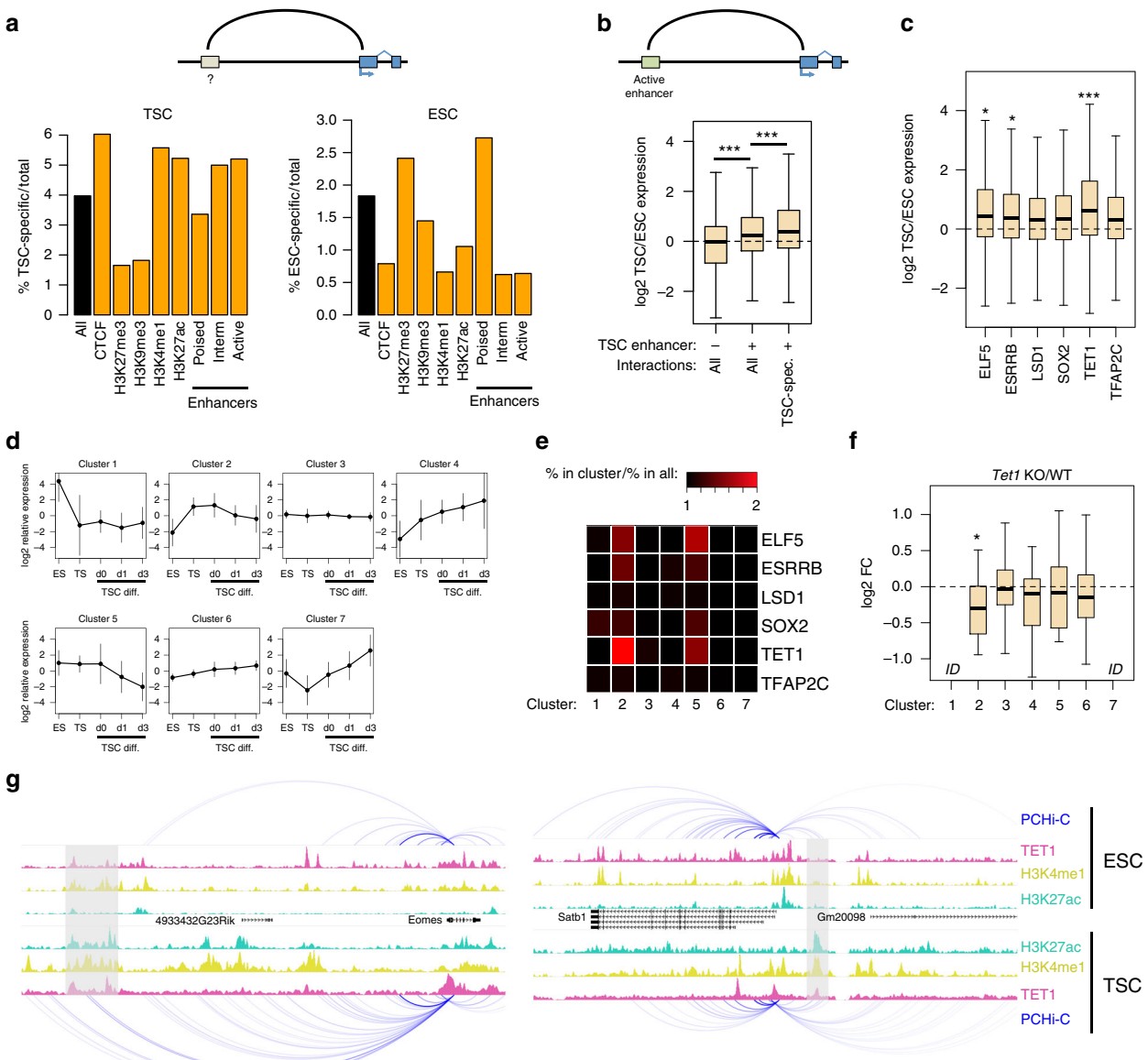

**Fig. 4** TSC-specific interactions involve TET1-regulated genes. **a** Percentage of cell-specific interactions among all interactions (black bars) or promoter–genome interactions with the indicated proteins/modifications/features (orange bars). **b** Expression ratio (log2) between TSCs and ESCs for genes involved in interactions (all or ESC-specific) with active enhancers, and compared with genes that do not interact with any active enhancer. ***$p < 0.0005$ (t-test, corrected for multiple comparisons). **c** Expression ratio (log2) between TSCs and ESCs for genes involved in interactions with active enhancers bound by the indicated transcription factors in TSCs. *$p < 0.005$, ***$p < 0.00005$, compared to LSD1 (ANOVA with Tukey post hoc test). **d** Relative expression profiles of genes grouped by k-means clustering based on expression in ESCs, TSCs and differentiating TSCs (days of differentiation are indicated). Error bars represent standard deviations. **e** Heatmap displaying the enrichment of genes interacting with enhancers bound by the indicated transcription factors in each cluster (relative to the abundance across all clusters). **f** Expression ratio (log2) between *Tet1* KO and WT TSCs for genes interacting with enhancers bound by TET1 in each cluster. ID indicates insufficient data. *$p < 0.01$, compared to all genes (ANOVA with Tukey post hoc test). **g** Examples of TET1-regulated genes displaying interactions with TET1-bound enhancers in TSCs. Grey boxes highlight TET1-bound enhancers in TSCs. Boxplot midline represents median, box edges the first and third quartiles, and whisker edges are the last data points within 1.5× the interquartile range

interactions was TSC-specific (Fig. 4b). These results suggest that enhancer-driven gene activation in TSCs involves the action of both newly wired interactions and the emergence of enhancer activity at established interactions.

We next sought to link enhancer–gene interactions with binding of key TSC transcription factors and epigenetic modifiers. We therefore processed TSC ChIP-seq data for CDX2, ELF5, EOMES, ESRRB, LSD1, NR0B1, SOX2, TET1 and TFAP2C[13,14,34–36]. Out of these, ELF5, ESRRB, LSD1, SOX2, TET1 and TFAP2C had a large proportion (25–36%) of peaks

overlapping our stringent list of active enhancers (Supplementary Fig. 8b). We therefore focused on these transcription factors and epigenetic modifiers to evaluate their putative gene regulatory effect via enhancers. Interestingly, compared to LSD1-bound enhancers (which represent 54% of all active enhancers), TET1-bound enhancers were strongly associated with a higher TSC/ESC expression ratio, with ELF5 and ESRRB also displaying a significant but milder effect (Fig. 4c). To further dissect the association between TSC gene expression and transcription factor or epigenetic modifier action at enhancers, we analysed

transcriptomic data from differentiating TSCs[35] together with ESC and TSC data. We grouped genes into seven clusters based on their expression profile (Fig. 4d) and then asked which proteins are bound to enhancers interacting with these genes. Interestingly, genes in clusters 2 and 5, which display high expression in undifferentiated TSCs, were enriched for enhancers bound by multiple proteins (Fig. 4e). Particularly prominent was the enrichment of TET1-bound enhancers interacting with cluster 2 genes. As expected, TET1-bound enhancers were associated with DNA hypomethylation in TSCs (Supplementary Fig. 9). To test for a functional relationship between enhancer binding and gene regulation, we analysed transcriptomic data from *Esrrb* or *Elf5* knockdowns[14] and from *Tet1* knockout TSCs[36]. We restricted our analysis to genes interacting with enhancers bound by the respective transcription factor. In ELF5- and ESRRB-depleted cells, expression changes were not limited to genes in clusters 2 and 5, and generally followed a similar pattern to that seen in differentiated TSCs (Supplementary Fig. 8c), suggesting that gene de-regulation in these cells is directly and indirectly affected by transcription factor depletion. However, in *Tet1* KO cells there was a clear preference for downregulation of genes associated with cluster 2 (Fig. 4f), in line with the enriched binding of TET1 to enhancers controlling these genes.

Notable examples of TSC-relevant genes that were affected by *Tet1* KO and that interact with TET1-bound enhancers include *Eomes*, *Satb1* (Fig. 4g), *Itga7*, *Elf5*, *Fgfr2* and *Itgav*. Of particular note, *Eomes* establishes a TSC-specific interaction with a TET1-bound enhancer located 85 kb upstream of its promoter (Fig. 4g). These results suggest that TET1 modulates TSC-specific gene expression through regulation of enhancer activity, which partly requires stem cell type-specific chromatin folding.

## Discussion

Differentiation between the embryonic and extraembryonic lineages involves the concerted effort of signalling cues, cell-specific activation of transcription factors, and remodelling of the epigenetic landscape. Here we have shown that the first cell lineage decision is also accompanied by pronounced reshaping of the 3D chromatin conformation, which reflects and possibly reinforces the reprogramming barrier imposed by other layers of gene regulation. While our study was limited to stem cell line models, ESC Hi-C data strongly resemble that from ICM cells and post-implantation epiblast[19,37], and the TSC gene regulatory landscape has been extensively validated in vivo (see e.g. refs. [1,23,38]).

The chromatin folding features that differentiate ESCs and TSCs are: (1) an enrichment for repressive interactions in ESCs, mainly between gene promoters, but also involving H3K27me3-marked enhancers and (2) an enrichment for active enhancer–gene interactions in TSCs. Similar observations have recently been made during neuronal differentiation of ESCs, wherein PRC1-associated interactions are disrupted and enhancer–gene contacts become progressively more abundant[31]. Increased enhancer connectivity is also seen in foetal liver cells when compared with ESCs[24]. These different 3D chromatin arrangements probably reflect a switch from a transcriptionally poised state of developmental genes in pluripotent ESCs to the activation of specific subsets of these genes in more developmentally restricted lineages. However, unlike the neuronal lineage, the trophoblast lineage is not derived from the pluripotent state represented by ESCs, which does not have the potential to form extraembryonic tissues. An obvious question is therefore how the genome is spatially arranged in the lead up to the separation of these two lineages in the blastocyst. Recent work in preimplantation embryos has suggested that the 3D chromatin

architecture undergoes a progressive maturation during early development, starting out with weak topologically associating domains and a global depletion of long-range interactions after fertilisation that become progressively stronger or more prominent as development progresses[37]. However, it remains unclear how finer scale gene–gene and gene–enhancer contacts develop during this period and into the separation of the embryonic and extraembryonic lineages. Future work on this question could explore the use of single-cell Hi-C technology[39] to profile individual blastomeres as they acquire expression of key TSC markers such as CDX2.

We have found that homotypic interactions between H3K27me3-marked loci in ESCs are associated with the repression of TSC-expressed genes. Interestingly, although virtually all of the genes involved in these interactions are devoid of H3K27me3 in TSCs (Fig. 2b), only differentially interacting genes display higher expression in TSCs (Fig. 2d). The same holds true when analysing PRC1-deficient ESCs, suggesting that expression of TSC-specific genes is particularly affected by the loss of PRC1-associated interactions. Repressive chromatin conformations may therefore represent a critical layer in silencing trophoblast genes in the embryonic lineage. Accordingly, TSCs are globally depleted for H3K27me3 (ref. [20]) and extraembryonic tissues are far less affected than embryonic tissues in mutants of the PRC2 catalytic subunit *Ezh2* (ref. [40]). H3K27me3 deposition increases throughout preimplantation development, with the majority of bivalent domains being established during the morula-to-blastocyst transition[38], but it remains to be seen how this is integrated by the 3D arrangement of the genome to enable control of trophoblast-specific genes throughout this period. Notably, long-range interactions between PcG-bound regions similar to the ones we detected here in primed (serum-grown) ESCs are not present in ground-state (2i-grown) pluripotent ESCs[41]. Thus, PcG-dependent chromosomal interactions appear to be established in post-implantation pluripotent stem cells, and are then disrupted during cell fate specification during development[31], most likely facilitating the activation of key germ layer genes.

We also found that TSCs bear a higher proportion of cell-specific interactions between genes and active enhancers when compared to ESCs. Our data suggest that the key TSC transcription factors ELF5 and ESRRB may play particularly important roles in the enhancer-driven regulation of TSC gene expression. Additionally, the DNA methylation oxidising enzyme TET1 also seems to play a key role, similar to what has been observed at other tissue-specific enhancers[42]. Therefore, similar to ESRRB[14], TET1 is a factor shared by both ESCs and TSCs that gains context-specific functions, and our data suggest that the tissue-specific 3D conformation of the genome helps to enact such functions. Alternatively, it is the action of TET1 that modulates gene–enhancer contacts in TSCs. Interestingly, in gliomas with gain of function *IDH* mutations, interfering with TET function leads to aberrant gene expression, at least in part due to an increase in DNA methylation that impairs the binding of the methylation-sensitive chromosomal insulator protein CTCF[43]. DNA methylation of CTCF binding sites has also been shown to control enhancer–promoter interactions at the imprinted *H19* locus[44,45]. Furthermore, in concert with other epigenetic mechanisms, DNA methylation has been shown to facilitate[46] or weaken[47] enhancer–promoter contacts at specific loci. It is therefore likely that proteins involved in DNA (de)methylation direct enhancer–promoter communication not only by altering the methylation status at enhancers and promoters themselves, but also through the establishment or disruption of spatial chromosomal domains that dictate the access of regulatory sequences to target genes. The precise role of DNA methylation in shaping ESC genome topology and promoter–enhancer

interactions to maintain stem cell identity and to protect against trans-differentiation into TSCs[1] remains a subject for future studies.

Taken together, we describe here the distinctly different 3D chromatin organisation in TSCs compared to ESCs and identify a highly TSC-specific set of TET1-bound enhancers that drive expression of key TSC genes.

## Methods

**Cell culture.** Chimera-competent TS-EGFP cells (mixed ICR×129 background; kind gift from Dr. Janet Rossant)[6] were cultured under routine conditions (20% foetal bovine serum, 1 mM Na-pyruvate, Pen/Strep, 50 μM 2-mercaptoethanol, 25 ng/ml bFGF and 1 μg/ml heparin in RPMI1640, with 70% of the medium pre-conditioned on embryonic feeder cells). J1 ESCs (129S4/SvJae; ATCC SCRC-1010)[48] were expanded on irradiated primary embryonic fibroblasts under standard pluripotent conditions (15% foetal bovine serum) on tissue culture plates coated with 0.1% gelatin. To harvest the cells and remove contaminating feeder cells, ESCs were trypsinized and feeders allowed to re-attach for 30 min before collecting non-attached ESCs—this procedure was performed twice.

**Promoter capture Hi-C.** PCHi-C experiments were performed in duplicate for each cell line. Thirty to 40 million ESCs or TSCs were fixed in 2% formaldehyde (Agar Scientific) for 10 min at room temperature, and Hi-C libraries were prepared as described[49], with minor modifications. Briefly, crosslinked chromatin was digested with HindIII (NEB), and restriction fragment ends were labelled using biotin-14-dATP (Life Technologies) and DNA Polymerase I (Large Klenow Fragment; NEB). After ligation in nuclei[50] (5 h at 16 °C with T4 DNA ligase; Life Technologies), the crosslinks were reversed by Proteinase K digest (65 °C overnight), and the DNA was purified using phenol/chloroform (Sigma-Aldrich) extraction. After removal of biotin from unligated DNA ends, the DNA was sonicated (Covaris E220) to an average size of around 400 base pairs, and end-repaired using DNA Polymerase I (Large Klenow Fragment), T4 DNA polymerase, and T4 DNA polynucleotide kinase (all NEB). dATP was added to the 3′ ends of the DNA (using Klenow Fragment (3′ → 5′ exo−); NEB), and the DNA was subjected to double-sided SPRI bead size selection (AMPure XP beads; Beckman Coulter). Biotin-marked ligation products were isolated using MyOne Streptavidin C1 Dynabeads (Life Technologies), and after adapter ligation (Illumina PE adapter with T4 DNA ligase; NEB) the bead-bound Hi-C DNA was amplified with seven PCR amplification cycles using PE PCR 1.0 and PE PCR 2.0 primers (Illumina).

Promoter Capture Hi-C was performed as described[24], using a custom-made RNA capture bait system (Agilent Technologies) consisting of 39,021 individual biotinylated RNAs targeting the ends of 22,225 promoter-containing mouse HindIII restriction fragments. The Hi-C library DNA (500–750 ng) was mixed with hybridisation blockers (Agilent Technologies) and denatured for 5 min at 95 °C, then incubated with hybridisation buffer and the RNA capture bait system at 65 °C for 24 h (in a MJ Research PTC-200 PCR machine). After the hybridisation incubation, DNA/biotin-RNA was isolated using MyOne Streptavidin T1 Dynabeads (Life Technologies), following the manufacturer's instructions (SureSelect Target Enrichment; Agilent Technologies). After the final wash in wash buffer 2 (Agilent Technologies), the beads were resuspended in 300 μl NEBuffer 2 (NEB), isolated on a DynaMag-2 magnet (Life Technologies), and resuspended in a final volume of 30 μl NEBuffer 2. After a post-capture PCR (four amplification cycles using Illumina PE PCR 1.0 and PE PCR 2.0 primers), the Promoter CHi-C libraries were purified with AMPure XP beads (Beckman Coulter) and paired-end sequenced (HiSeq 1000, Illumina) at the Babraham Institute Sequencing Facility.

**PCHi-C data processing.** Raw sequencing reads were processed using the HiCUP pipeline (v0.5.8), which maps the ditags against the mouse genome (mm10), filters experimental artefacts, such as circularised reads and religations, and removes duplicate reads[51]. For read yields and quality control metrics see Supplementary Fig. 1. To maintain the statistical power even between the two cell types, reads from the TSC dataset were randomly subset to match the number of reads in the ESC dataset.

Significant promoter–genome interactions were called using the GOTHiC BioConductor package (v1.14.0). To exclude low frequency contacts, which are likely to be non-functional or have a role only in a very small proportion of cells, we established stringent q-value thresholds that ensured only visibly strong contacts with a given promoter were kept—based on empirical assessment of 4C-like tracks. To formally define this threshold, for each promoter we first plotted the cumulative distribution of the significance levels ($-\log 10$ q-value) of its interactions. We then used the inflection point of the first derivative of this cumulative significance curve to define a promoter-specific significance threshold. To take into account differences in biases affecting promoter–promoter interactions, these were called with GOTHiC using a modified logit background distribution as described in ref. [24].

Significant differential interactions were identified by a modified version of the GOTHiC pipeline, where two conditions are directly compared and the expected values are the number of reads observed in the reference condition. For scaling, the number of total captured reads per condition is used. To enable logarithmic transformation of data, for those interactions where no reads were detected for one of the cell types, we added 1 in the cell type with the missing value. For multiple testing correction we use Independent Hypothesis Weighting (IHW v.1.6.0) and a false discovery rate of 5%.

For the sole purpose of comparing our interactions with CHiCAGO-processed PCHi-C data from human blood cell types[30] (Supplementary Data 1) we also identified ESC and TSC interactions using the CHiCAGO Bioconductor package (v1.6.0)[52]. Interactions were merged from biological replicates using CHiCAGO and those with score >5 were considered in our analysis.

For Ring1A KO and Ring1A/B DKO PCHi-C data[27] (Supplementary Data 1), processed GOTHiC calls were downloaded and lifted over from mm9 to mm10. For genome-wide ESC Hi-C data[31] (Supplementary Data 1), Knight-Ruiz normalised matrices for regions of interest were extracted using Juicer[53].

**ChIP-seq and DNase-seq data processing.** Fastq files from publicly available datasets (Supplementary Data 1) were downloaded via the EMBL-EBI European Nucleotide Archive. Reads were trimmed using Trim_galore! and aligned to the mm10 genome assembly using Bowtie2 v2.1.0 (ref. [54]), followed by filtering of uniquely mapped reads. ChIP-seq peak detection was performed using MACS2 v2.1.1 (ref. [55]) with $-q$ 0.05; for histone marks the option --broad was used. DNase-seq peak detection was performed using F-seq v1.84 (ref. [56]) with options $-f$ 0 $-t$ 6.

ChIP-seq signal densities were calculated by normalising read counts at each peak by the total read counts and peak length. The MatchIt R library was used to identify matched peaks (in length and ChIP-seq signal density) between ESC-specific interactions and shared interactions.

Enhancers were first defined based on the H3K4me1 peaks in each cell type: (1) H3K4me1 peaks overlapping H3K27ac but not H3K27me3 were classified as active enhancers, (2) those overlapping H3K27me3 as poised and (3) those overlapping neither mark as intermediate. Peaks within 1 kb of RefSeq transcriptional start sites were excluded. A second list of active enhancers in TSCs was defined by finding the intersection between DNase I, H3K4me1 and H3K27ac peaks, merging enhancers within 300 bp of each other, and filtering out particularly small (<100 bp) or large (>3 kb) enhancers.

**RNA-seq data processing.** Fastq files from publicly available datasets (Supplementary Data 1) were downloaded via the EMBL-EBI European Nucleotide Archive. Reads were trimmed using Trim_galore! and aligned the mm10 genome assembly with Tophat v2.0.9 (ref. [57]) using a transcriptome index from Illumina's iGenomes. For Ring1A KO and Ring1A/B DKO RNA-seq, mapped bam files were directly downloaded from the respective data repository. Transcript FPKM (fragments per kilobase of transcript per million mapped reads) values were extracted using Seqmonk.

To exclude low-expressing, high-variance transcripts when evaluating differences in expression, a minimum expression threshold was employed based on the distribution of log2 FPKM values ($-3$ for Ring1 KO data, $-1$ for the remaining). K-means clustering of expression data was performed using this minimum expression threshold, with the number of clusters ($n = 7$) decided empirically.

**Code availability.** Custom scripts used to analyse the data in this study are available from the authors upon request. The differential interaction calling algorithm will be included in the next release of GOTHiC, available from Bioconductor (http://bioconductor.org).

## Data availability
The PCHi-C data that support the findings of this study have been deposited in ArrayExpress with the accession code E-MTAB-6585.

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

## Acknowledgements

We thank Sarah Elderkin for ESC culture, Kristina Tabbada and Clare Murnane for next-generation sequencing; Steven Wingett for initial PCHi-C data processing; Boyan Bonev for help with extracting Hi-C data and Sarah Burge for useful discussions. S.S. was supported by the Biotechnology and Biological Sciences Research Council (BB/J004480/1) and a Career Progression Fellowship from the Babraham Institute. B.M. holds an MRC eMedLab Medical Bioinformatics Career Development Fellowship, funded from award MR/L016311/1. C.D.T. was supported by The Medical College of Saint Bartholomew's Hospital Trust. M.R.B. is a Sir Henry Dale Fellow (101225/Z/13/Z), jointly funded by the Wellcome Trust and the Royal Society. This project was enabled through access to the MRC eMedLab Medical Bioinformatics infrastructure, award MR/L016311/1.

## Author contributions

M.R.B., M.H. and S.S. designed the study and experiments. C.E.S. performed cell culture. S.C. generated the *Tet1* KO TSC line. S.S. performed PCHi-C experiments. M.R.B., B.M.

and C.D.T. performed computational analyses. B.M. and E.D. developed the differential interaction calling algorithm. M.R.B. and M.H. wrote the manuscript with all other authors.

## Additional information

**Competing interests:** The authors declare no competing interests.

