## [Peer Review File · Nature Communications]

Reviewer #1 (Remarks to the Author):

In this paper, Schoenfeld et al. describe H3K27me3-associated and active chromatin interactions in embryonic stem cells and trophoblast stem cells. The main finding is that genes that are expressed in trophoblast but silenced in embryonic stem cells are associated with repressive chromatin interactions in embryonic stem cells. These are dependent on PRC1 and associated with TET. If true, these claims are novel and will add to our understanding of chromatin interactions in lineage specification. Also, in general, the paper is well written and the figures are well made. I just have the following major comments:

1. My most major concern is that the TSC and ESC cells are not from the same source (i.e. the TSC were not derived from the ESC) as can be shown from the sex differences. How can we be sure that the differences seen are truly due to the differences between TSC and ESC, and are not due to differences between different individuals? Similarly, are the DKO cells derived from different sources too?

2. The main finding is that genes that are expressed in trophoblast but silenced in embryonic stem cells are associated with repressive chromatin interactions in embryonic stem cells. Is the converse true? Are genes expressed in embryonic stem cells but silenced in trophoblast associated with repressive chromatin interactions in trophoblast?

3. Was pHi-C also done in TET1 knockout trophoblast cells, to look at changes in chromatin interactions in such cells?

4. It is unclear from the materials and methods how the Ring1A KO and Ring1A/B

DKO cells were prepared for pHi-C.

5. In evaluating differences between pHi-C datasets it is important to understand how similar are the replicates. Supplementary 4A shows some examples of repressive chromatin interactions, with rep 1 and rep 2 of the pHi-C data shown. It would be good if some examples of repressive chromatin interactions that are different between the DKO cells and the wild-type cells can also be shown similarly, so that claims about the reproducibility and clear differences between the chromatin interactions in the pHi-C of the DKO and the wild-type cells can be manually inspected.

6. The authors say in the discussion "Interestingly, the absence of these interactions in TSCs is a better predictor of gene expression than the differential deposition of H3K27me3." I tried looking in the results section but could not find evidence for this having been discussed in the results section. Where is this shown?

Reviewer #2 (Remarks to the Author):

In this report, the authors investigate repressive and active chromatin interactions in mouse embryonic stem (ES) cells and mouse trophoblast stem (ES) cells. Promoter capture HiC analyses were performed in cultured ES and TS cells and layered on existing transcriptional profiles and ChIPseq analyses for transcription factor binding sites and other proteins interacting with the genome. Interesting chromatin interactions were seen for ES cells versus TS cells, especially related to homotypic interactions for H3K27me3 and polycomb proteins and TET-1 interactions on ES cell versus TS cell promoters. The work represents an important contribution. Some concerns with the experimentation and interpretation are presented below.

1. The authors should provide some additional detail about the TS cells used for their analyses, including genetic background. The rationale for using a genetically manipulated TS cell line overexpressing EGFP for their analysis is not apparent. Was the mouse ES cell line also manipulated to overexpress EGFP? Inclusion of a TS cell line overexpressing EGFP was not necessary and could potentially confound some of the results.

2. The results are potentially intriguing. However, a significant concern is the relevance of the cell culture experiments with mouse ES and TS cells to in vivo mouse embryonic development. The authors do not provide any in vivo validation of their data generated with ES and TS cell lines. This represents a limiting feature of the report.

3. Page 10, lines 253-257. Some confusion: LSD1 and TET1 are not generally considered to be defined as transcription factors.

Response to reviewers

Reviewer #1 (Remarks to the Author):

In this paper, Schoenfeld et al. describe H3K27me₃-associated and active chromatin interactions in embryonic stem cells and trophoblast stem cells. The main finding is that genes that are expressed in trophoblast but silenced in embryonic stem cells are associated with repressive chromatin interactions in embryonic stem cells. These are dependent on PRC1 and associated with TET. If true, these claims are novel and will add to our understanding of chromatin interactions in lineage specification. Also, in general, the paper is well written and the figures are well made. I just have the following major comments:

1. My most major concern is that the TSC and ESC cells are not from the same source (i.e. the TSC were not derived from the ESC) as can be shown from the sex differences. How can we be sure that the differences seen are truly due to the differences between TSC and ESC, and are not due to differences between different individuals? Similarly, are the DKO cells derived from different sources too?

We appreciate the reviewer's concern. To clarify, *bona fide* TSCs are not derived from ESCs, but directly from *in vivo* tissue (either E3.5 trophectoderm or E6.5 extraembryonic ectoderm). To the best of our knowledge, all commonly used ESC and TSC lines have been independently derived from different embryos. We used well-characterised cell lines (J1 ESCs and TS-EGFP) for which abundant epigenomic and transcriptomic data are available, which we used here. This enabled us to attain a deeper understanding of the molecular determinants underlying differences in genome folding between ESCs and TSCs.

The differences we observed in genome folding are very unlikely due to differences between embryos, as patterns seen in J1 ESCs (Figure 2E,F) are also seen in at least two other independently derived ESC lines: 1) Oct4GiP ESCs (Supplementary Figure S4), which were used by Bonev et al. (PMID: 29053968), and 2) Ring1A KO ESCs (Figure 3D), which were used by Schoenfelder et al. (PMID: 26323060). The different genome topology of TSCs therefore reflects their distinct cellular identity.

The Ring1A/B DKO cells are derived from the Ring1A KO cells, which harbor floxed Ring1B alleles (PMID: 15525528). Ring1B is deleted upon tamoxifen treatment. They are therefore directly comparable with each other. We have made this clearer in the main text (page 8).

2. The main finding is that genes that are expressed in trophoblast but silenced in embryonic stem cells are associated with repressive chromatin interactions in embryonic stem cells. Is the converse true? Are genes expressed in embryonic stem cells but silenced in trophoblast associated with repressive chromatin interactions in trophoblast?

The reviewer poses an interesting question. However, TSC-specific interactions are rarely associated with H3K27me3 in TSCs, involving only 5 genes, which does not allow us to draw general conclusions about the potential impact on expression. We have added this information to the results section (page 6).

3. Was pHi-C also done in TET1 knockout trophoblast cells, to look at changes in chromatin interactions in such cells?

We have not performed these experiments because it has previously been shown that DNA methylation has no impact on genome topology (PMID: 29162810). TET enzymes are thought to modulate the accessibility of methylation-sensitive transcription factors to enhancers, but are unlikely to affect gene-enhancer contacts, which rely on structural proteins.

4. It is unclear from the materials and methods how the Ring1A KO and Ring1A/B DKO cells were prepared for pHi-C.

To clarify, we only used processed PChi-C data from a previous publication (PMID: 26323060), which is why we do not describe details about those experiments in this manuscript and instead refer to the original publication.

5. In evaluating differences between pHi-C datasets it is important to understand how similar are the replicates. Supplementary 4A shows some examples of repressive chromatin interactions, with rep 1 and rep 2 of the pHi-C data shown. It would be good if some examples of repressive chromatin interactions that are different between the DKO cells and the wild-type cells can also be shown similarly, so that claims about the reproducibility and clear differences between the chromatin interactions in the pHi-C of the DKO and the wild-type cells can be manually inspected.

We appreciate the suggestion and have included a new Supplementary Figure S6 displaying 4C-like tracks of individual replicates of the KO/DKO PChi-C data.

6. The authors say in the discussion "Interestingly, the absence of these interactions in TSCs is a better predictor of gene expression than the differential deposition of H3K27me3." I tried looking in the results section but could not find evidence for this having been discussed in the results section. Where is this shown?

We apologise if this was not clear. We have rephrased this sentence and made references to the respective figures, as such: "Interestingly, although virtually all of the genes involved in these interactions are devoid of H3K27me3 in TSCs (Figure 2B), only differentially interacting genes display higher expression in TSCs (Figure 2D)".

Reviewer #2 (Remarks to the Author):

In this report, the authors investigate repressive and active chromatin interactions in mouse embryonic stem (ES) cells and mouse trophoblast stem (TS) cells. Promoter capture HiC analyses were performed in cultured ES and TS cells and layered on existing transcriptional profiles and ChIPseq analyses for transcription factor binding sites and other proteins interacting with the genome. Interesting chromatin interactions were seen for ES cells versus TS cells, especially related to homotypic interactions for H3K27me3 and polycomb proteins and TET-1 interactions on ES cell versus TS cell promoters. The work represents an important contribution. Some concerns with the experimentation and interpretation are presented below.

1. The authors should provide some additional detail about the TS cells used for their analyses, including genetic background. The rationale for using a genetically manipulated TS cell line overexpressing EGFP for their analysis is not apparent. Was the mouse ES cell line also manipulated to overexpress EGFP? Inclusion of a TS cell line overexpressing EGFP was not necessary and could potentially confound some of the results.

We appreciate the reviewer's concern. The main reason to choose the TS-EGFP line was their competency to generate extraembryonic chimeras *in vivo*, which is well established and recently confirmed by us (PMID: 25423963). This demonstrates the functional resemblance of TS-EGFP cells to the early trophoblast lineage. Additionally, there is an abundance of transcriptomic and epigenomic data for this line, which we used here.

TS-EGFP cells were derived by Tanaka et al. (PMID: 9851926) from B5/EGFP transgenic mice (PMID: 9867352), which are of mixed ICR x 129 background. Note that the EGFP manipulation occurred ahead of the derivation of the TSC line.

We have modified the methods section to highlight the chimera competency of the cells and include genetic background information (page 14).

2. The results are potentially intriguing. However, a significant concern is the relevance of the cell culture experiments with mouse ES and TS cells to *in vivo* mouse embryonic development. The authors do not provide any *in vivo* validation of their data generated with ES and TS cell lines. This represents a limiting feature of the report.

The reviewer raises an important point. We agree that stem cell line models are only useful as long as they reflect the *in vivo* state. It has indeed been previously demonstrated that Hi-C data from ESCs resemble that of the inner cell mass and of the post-implantation epiblast (PMID: 28703188, 29203909). We have now pointed this out in the discussion (page 12). To assess whether this resemblance applied to our findings, we analysed one of these published datasets and found evidence that the H3K27me3-associated interactions seen Figure 2E,F are also present in ICM:

However, these data are of markedly lower resolution/power compared to our PChi-C data and we would therefore not feel comfortable including this information in the manuscript. Indeed, C-type of experiments (e.g., 3C, 4C, Hi-C) in small cell numbers are in their infancy and are extremely challenging from a technical perspective. This is exacerbated in the case of promoter capture Hi-C, which requires a larger number of cells to ensure library complexity.

We chose to use TSCs as they have been instrumental in uncovering the molecular underpinnings of *in vivo* trophoblast biology. Importantly, these cells readily make extraembryonic chimeras when injected into blastocysts. Moreover, there is abundant evidence that demonstrate their resemblance to the *in vivo* state at the gene regulatory level. This includes: 1) the roles of transcription factors such as ELF5 (PMID: 18836439, 26584622), CDX2 (PMID: 16325584, 25423963), ETS2 (PMID: 17977525) and GATA3 (PMID: 19700764), 2) the distribution of active and inactive histone modifications (PMID: 20573702, 27626379), and 3) key DNA methylation differences between ESCs and TSCs (PMID: 18836439, 23034951, 26812015). The PChi-C data generated here therefore most likely reflects these other aspects of TSC gene regulation and resembles the *in vivo* state. Nonetheless, we fully accept the limitations of cell line models and have highlighted this in the discussion (page 12).

3. Page 10, lines 253-257. Some confusion: LSD1 and TET1 are not generally considered to be defined as transcription factors.

We thank the reviewer for pointing this out. We have modified this section to refer to “transcription factors and epigenetic modifiers” instead of transcription factors alone.

Reviewer #1 (Remarks to the Author):

In this review I will refer mainly to the point by point reply from the authors.

1. Reply by the authors to item 1: The second paragraph of the author's response helps to assuage my concern about the different origins of the cell lines. The authors should include such an analysis in the paper in one of the Supplementary Figures.

2. Reply by the authors to item 3: The authors say "it has previously been shown that DNA methylation has no impact on genome topology" - there seems to be some debate in the field about this question. CTCF and cohesin cannot bind easily to methylated DNA, and in the case of PMID: 26700815 they argue that the oncometabolite leads to alterations in DNA methylation which lead to changes in TAD boundaries. I agree that PMID: 29162810 does say that there methylation is not required for the formation of A/B compartments, but compartments are quite large and may not be controlled by the same mechanisms as TADs and individual enhancer-promoter chromatin interactions, which are the ones explored but the authors in this paper. However, the methylation analyses can be left for future exploration.

I have no further questions or concerns.

Reviewer #2 (Remarks to the Author):

The authors have satisfactorily addressed my concerns

Response to reviewers

Reviewer #1 (Remarks to the Author):

In this review I will refer mainly to the point by point reply from the authors.

1. Reply by the authors to item 1: The second paragraph of the author's response helps to assuage my concern about the different origins of the cell lines. The authors should include such an analysis in the paper in one of the Supplementary Figures.

We have added an analysis of a genome-wide Hi-C dataset generated from an different ESC line (Supplementary Figure 2b). From these data we extracted normalized read counts for the ESC- or TSC-specific interactions that we identified in our cell line. The data clearly show that ESC-specific contacts are stronger than TSC-specific ones in the published dataset, arguing that the differential interactions detected reflect distinct cellular identities.

2. Reply by the authors to item 3: The authors say "it has previously been shown that DNA methylation has no impact on genome topology" - there seems to be some debate in the field about this question. CTCF and cohesin cannot bind easily to methylated DNA, and in the case of PMID: 26700815 they argue that the oncometabolite leads to alterations in DNA methylation which lead to changes in TAD boundaries. I agree that PMID: 29162810 does say that there methylation is not required for the formation of A/B compartments, but compartments are quite large and may not be controlled by the same mechanisms as TADs and individual enhancer-promoter chromatin interactions, which are the ones explored but the authors in this paper. However, the methylation analyses can be left for future exploration.

We thank the reviewer for highlighting these findings, which raise interesting questions. As a result, we have added a discussion on this topic to the manuscript (page 14).